# Pathophysiological Roles of Ion Channels in Epidermal Cells, Immune Cells, and Sensory Neurons in Psoriasis

**DOI:** 10.3390/ijms25052756

**Published:** 2024-02-27

**Authors:** Hyungsup Kim, Mi Ran Choi, Seong Ho Jeon, Yongwoo Jang, Young Duk Yang

**Affiliations:** 1Brain Science Institute, Korea Institute of Science and Technology (KIST), Seoul 02792, Republic of Korea; suppy88@kist.re.kr; 2Laboratory Animal Research Center, Ajou University School of Medicine, Suwon 16499, Republic of Korea; mrchoi2007@ajou.ac.kr; 3Department of Pharmacy, College of Pharmacy and Institute of Pharmaceutical Sciences, CHA University, Pocheon 11160, Republic of Korea; cmb_jsh@naver.com; 4Department of Pharmacology, College of Medicine, Hanyang University, Seoul 04736, Republic of Korea

**Keywords:** psoriasis, calcium, ion channels, epidermal cell, immune cell, sensory neuron

## Abstract

Psoriasis is a chronic inflammatory skin disease characterized by the rapid abnormal growth of skin cells in the epidermis, driven by an overactive immune system. Consequently, a complex interplay among epidermal cells, immune cells, and sensory neurons contributes to the development and progression of psoriasis. In these cellular contexts, various ion channels, such as acetylcholine receptors, TRP channels, Ca^2+^ release-activated channels, chloride channels, and potassium channels, each serve specific functions to maintain the homeostasis of the skin. The dysregulation of ion channels plays a major role in the pathophysiology of psoriasis, affecting various aspects of epidermal cells, immune responses, and sensory neuron signaling. Impaired function of ion channels can lead to altered calcium signaling, inflammation, proliferation, and sensory signaling, all of which are central features of psoriasis. This overview summarizes the pathophysiological roles of ion channels in epidermal cells, immune cells, and sensory neurons during early and late psoriatic processes, thereby contributing to a deeper understanding of ion channel involvement in the interplay of psoriasis and making a crucial advance toward more precise and personalized approaches for psoriasis treatment.

## 1. Introduction

Psoriasis, which affects 125 million people—2 to 3 percent of the total population worldwide—is a chronic inflammatory skin disease [1,2]. Psoriasis causes considerable emotional, physical, and social burdens that lower the quality of life for most people who have it. Psoriasis presents as red and scaly plaques that occur most commonly on the scalp, knees, elbows, and lower back [3]. Psoriasis is characterized by hyperplasia of the epidermis through the rapid abnormal growth and differentiation of keratinocytes. Its pathophysiology is multifactorial, including an overactive immune system, pathophysiological signals triggered by environmental factors, and changes in ion channels [3,4,5]. Ion channels are transmembrane proteins that play a critical role in a variety of cellular processes, including the regulation of membrane potential, intracellular signaling, and cellular homeostasis [6]. Recently, it has been suggested that ion channels might also play a role in the pathogenesis of psoriasis.

The immunopathological features and functions of ion channels in psoriasis affect keratinocytes, immune cells, and dendritic cells, including in the peripheral nervous system [4,7,8]. The hyperproliferation of keratinocytes stimulates the activation of dendritic cells and immune cells, and those activated cells secrete pro-inflammatory cytokines such as interleukin (IL)-17, IL-22, tumor necrosis factor (TNF)-α, IL-23, and IFN-γ [5,7,9,10]. Those increased pro-inflammatory cytokines are responsible for the development of psoriasis.

Studies have suggested that ion-channel functional changes in keratinocytes could be an important factor in the pathogenesis of psoriasis [4,9,11]. In this review, we examine the ion channels in keratinocytes and their pivotal role in the pathogenesis of psoriasis, particularly focusing on two major types: nicotinic acetylcholine receptors (nAChRs) and transient receptor potential (TRP) channels. This review also considers the influence of tobacco-derived compounds, such as nicotine, on nAChRs, emphasizing the significance of α7nAChR in epidermal keratinocytes and its effects on the skin barrier function. On the other hand, the exploration of TRP channels, specifically TRPA1, TRPV1, TRPV3, and TRPV4, elucidates their involvement in various physiological processes within keratinocytes, ranging from skin barrier maintenance to inflammation and itch sensation [12,13,14,15]. Notably, the potential therapeutic implications of targeting these ion channels for psoriasis treatment are highlighted, offering insights into promising avenues for future research and intervention strategies.

Ion channels play a pivotal role in the intricate network of interactions between immune cells and skin cells, contributing to the pathogenesis of inflammatory conditions such as psoriasis. Among these channels, TRP channels, especially TRPA1 and TRPV1, show contrasting effects on CD4+ T cells in psoriatic inflammation. Whereas TRPA1 exacerbates psoriatic skin inflammation, TRPV1 enhances pro-inflammatory characteristics, suggesting that CD4+ T cell activity dynamically mediates inflammation [16,17]. In addition, calcium signaling, mediated by STIM1/ORAI1, emerges as a critical regulator of immune cell function, influencing neutrophil chemotaxis and presenting a potential target for psoriasis treatment. Calcium release–activated channels and arachidonic acid–regulated calcium-selective channels share similar biophysical properties, and their subunits regulate calcium in T cells [18,19,20]. On the other hand, α7nAChR mediates anti-inflammatory effects in immune cells, while voltage-gated potassium channels are involved in the proliferation, differentiation, and apoptosis of effector memory T cells [21,22,23]. Understanding the diverse roles of these ion channels in immune cells provides insight into the complex mechanisms that drive psoriatic inflammation and offers potential targets for therapeutic intervention.

Psoriasis, which is traditionally recognized as a skin disorder, also has influence on the peripheral nervous system, engaging ion channels in peripheral sensory neurons to mediate itch, pain, and immune responses. A previous study suggested that the nerve growth factor–mediated TRPV1 pathway is associated with the regulation of neurogenic pruritus, defined as an unpleasant sensation of the skin that provokes the urge to scratch [24]. In addition to the TRPV1 channel, other TRP channels, such as TRPV4 and TRPC4, also contribute to the occurrence of scratching behavior through peripheral neurotransmitter receptor–related pathways [5,25]. Therefore, understanding the complex interactions between ion channels and peripheral sensory neurons will provide insights for developing targeted therapeutic interventions against psoriasis.

This review explores the current literature to delineate the role of ion channels in the pathogenesis of psoriasis. Specifically, we examine the roles of calcium, potassium, sodium, and chloride channels in epidermal cells, immune cells, and sensory neurons in psoriasis (Figure 1, Table 1).

## 2. Ion Channels in Keratinocytes

### 2.1. Nicotinic Acetylcholine Receptors

A number of studies have demonstrated that skin exposure to tobacco-derived compounds can influence the homeostasis of epidermal keratinocytes, which is highly related to the pathological development of psoriasis. Indeed, calcium-permeable receptors and ion channels in epidermal keratinocytes respond to toxic tobacco-derived agents. The leaves of tobacco plants are primarily used for smoking in cigarettes and cigars and contain more than 7000 chemicals. Smoking tobacco produces a variety of harmful components in both the particulate and vapor phases that cause cancer and cardiovascular diseases in humans. In the particulate phase, nicotine is an alkaloid compound in tobacco that causes the addictive effects of tobacco products.

Nicotinic acetylcholine receptors (nAChRs) are ionotropic receptors that allow various cations to pass through the cellular membrane in response to acetylcholine, choline, and nicotine. Acetylcholine-responsive nAChRs are cholinergic receptors that mediate skeletal muscle contraction and transmit electrical signals between presynaptic and postsynaptic neurons within the sympathetic and parasympathetic nervous systems. Unlike acetylcholine-responsive muscarinic receptors, nAChRs are specifically activated by nicotine, which is how they got their name.

Among the nAChRs, α7nAChR is expressed in epidermal keratinocytes, sebocytes, and dermal fibroblasts of the skin. In general, α7nAChR is an ionotropic receptor with high Ca^2+^ permeability. In fact, nicotine induces an intracellular calcium influx in both human keratinocytes and the skin of α7nAChR-deficient mice, and that influx can be blocked by the application of α-bungarotoxin or antisense oligonucleotides [41]. Phenotypically, delayed epidermal turnover was observed in the epidermal cells of α7nAChR-deficient mice [41]. Another study reported that the mRNA and protein levels of pro-apoptotic Bad and Bax are reduced in the keratinocytes of α7nAChR-deficient mice [42]. Additionally, α7nAChR regulates the expression of zona occludens-1, also known as tight junction protein-1. Notably, exposure to nicotine resulted in a reduction in the barrier function associated with tight junctions, ultimately weakening the skins tight junction barrier [43].

Tobacco-derived compounds can also influence the toxicity of the epithelial cells (oral keratinocytes) that line the gingiva and esophagus [44,45]. Indeed, oral keratinocytes express nAChRs that can bind nicotine [46]. Exposing oral keratinocytes to nicotine was found to increase the mRNA and protein levels of the α3, α5, α7, β2, and β4 nAChR subunits by 1.5 to 2.9-fold [46]. Changes in nAChR expression were observed in both psoriasis mouse models and human skin, with a significant increase in α5 and α7 nAChR in psoriatic skin [4,11]. Notably, in palmoplantar pustulosis, a common type of psoriasis associated with smoking, α3 and α7 nAChRs are strongly expressed in sweat glands and ducts. Smoking has also been reported to increase the expression of α3 and α7 nAChR in sweat glands, contributing to palmoplantar pustulosis [47].

Targeting the regulation of nAChRs appears to hold promise as a therapeutic approach for psoriasis. Silencing α5nAChR inhibits keratinocyte proliferation and reduces psoriasis severity in α5nAChR-knockout mice by modulating the mitogen-activated protein kinase (MAPK)/nuclear factor kappa β (NF-κB) pathway [11]. The activation of nAChRs counteracts TNF-α-induced cutaneous inflammation by engaging the STAT3 and NF-κB signaling pathways, whereas treatment with an α7nAChR antagonist exacerbates psoriasis [4]. Agonists of α7nAChR, such as PNU-282987, AR-R17779, and tropisetron, have demonstrated efficacy in reducing inflammation by downregulating inflammatory factors and normalizing keratinocyte proliferation [4,48]

### 2.2. TRP Channels

#### 2.2.1. TRPA1

In the vapor phase of smoking tobacco, formaldehyde, acetaldehyde, and acrolein compounds contribute to tobacco’s harmful effects [49,50,51]. In fact, these volatile organic compounds activate Ca^2+^-permeable TRPA1 channels when they come into contact with the skin barrier. TRPA1 is a non-selective cation channel primarily expressed in sensory neurons, where it plays a critical role in conveying electrical signals related to pain, itch, temperature, and inflammation. Although TRPA1 was initially characterized as an ion channel primarily expressed in sensory neurons, growing evidence indicates that it is also functionally expressed in non-neuronal cells, including airway epithelial cells and epidermal keratinocytes. In airway epithelial cells, TRPA1 has plays a role in sensing and responding to various environmental irritants and pollutants, such as cigarette smoke, diesel exhaust particles, and volatile organic compounds. The activation of TRPA1 in these cells can lead to the release of inflammatory mediators and cytokines that contribute to respiratory symptoms. In epidermal keratinocytes, TRPA1 is involved in sensing skin irritants and noxious chemicals. Activation of TRPA1 in these cells releases pro-inflammatory factors that promote skin inflammation and hypersensitivity.

In this regard, Atoyan et al. reported the mRNA and protein expression of TRPA1 in various cell types isolated from human skin, including epidermal keratinocytes, dermal fibroblasts, and melanocytes [52]. Their results indicated that treatment with icilin, a TRPA1 agonist, significantly changes gene expression in primary cultured human keratinocytes. Specifically, the expression of genes involved in the control of keratinocyte proliferation and differentiation, such as bone morphogenetic protein 7, growth differentiation factor 15, heat-shock protein 27 (HSP27), HSP90, and regulatory factors stimulating keratinocyte differentiation, is altered upon icilin treatment. Additionally, icilin treatment increased the mRNA levels of pro-inflammatory interleukins (IL-1α and IL-1β) in keratinocytes, suggesting that TRPA1 has a potential role in inflammatory processes in the skin. They further found evidence that TRPA1 activation can accelerate the regeneration of the skin barrier when it is mechanically disrupted. A topical application of TRPA1 agonists such as isothiocyanate and cinnamaldehyde, as well as cold exposure for 1 min, facilitated skin barrier recovery after tape stripping on the intact skin of hairless mice. This effect was blocked by the TRPA1 inhibitor HC030031, indicating that TRPA1 activation was responsible for the accelerated skin barrier regeneration.

A study made an interesting observation about the response of keratinocytes to low temperature [53]. Around 60% of undifferentiated keratinocytes showed an increase in intracellular Ca^2+^ at approximately 17 °C, whereas only 10% of differentiated keratinocytes responded to low temperatures. Surprisingly, despite the difference in responsiveness to temperature, there was no significant variation in the mRNA and protein levels of TRPA1 between differentiated and undifferentiated keratinocytes. It was hypothesized that undifferentiated keratinocytes in the deeper layers of the epidermis might display greater sensitivity to low-temperature exposure than the differentiated keratinocytes located in the upper layers [54,55]. The specific factors or signaling pathways that govern this differential sensitivity of keratinocytes to low temperatures require further elucidation.

Although TRPA1 activation is associated with beneficial effects such as skin barrier regeneration, it is also implicated in inflammation-induced keratinocyte damage. In cultured HaCaT keratinocytes, TRPA1 expression can be upregulated in response to various pro-inflammatory stimuli, including cytokines such as TNF. The upregulation of TRPA1 by TNF is mediated through several signaling pathways, including the NF-κB, p38 MAPK, and c-Jun *N*-terminal kinase MAPK pathways [56]. Further investigation revealed that TRPA1 plays a role in generating chemokine monocyte chemoattractant protein-1 in TNF-induced HaCaT cells and mouse skin biopsies [56]. Indeed, it is intriguing that the expression of TRPA1 can be significantly reduced by the application of certain pharmacological drugs commonly used to treat skin inflammation. Two classes of drugs, calcineurin inhibitors (such as tacrolimus and cyclosporine) and the glucocorticoid dexamethasone, attenuate TRPA1 expression [56]. In addition, the plant polyphenol resveratrol (3,4,5-trihydroxystilbene) was found to suppress the increased reactive oxygen species (ROS) and carbonyl group adducts caused by exposing cultured HaCaT cells to cigarette smoke [57]. Interestingly, cigarette smoke was found to increase the mRNA and protein expression of TRPA1 in keratinocytes, an effect that was reduced by pretreatment with resveratrol [57]. Recent studies on TRPA1 in psoriasis have shed light on its involvement in the pathogenesis of the disease [15]. In human psoriatic lesion skin, TRPA1 was upregulated, indicating its potential role in driving the inflammatory process associated with psoriasis [15]. To investigate further, researchers used TRPA1-deficient mice and observed the effects of imiquimod (IMQ), a topical cream commonly used to induce psoriasis-like symptoms in animal models [15]. Interestingly, the TRPA1-deficient mice showed a significant reduction in IMQ-induced psoriatic symptoms, including skin barrier defects and the production of inflammatory mediators, suggesting that TRPA1 plays a crucial role in mediating the pathologic inflammation seen in IMQ-induced psoriasis. Those findings indicate that TRPA1 could be a promising target for new therapeutics to treat psoriasis.

Unlike human keratinocytes, murine keratinocytes express TRPA1 at low levels and with variants [26]. Remarkably, Zappia et al. demonstrated that mice lacking TRPA1 in their keratinocytes exhibited impaired mechanosensitivity to noxious and gentle mechanical stimuli, though their thermal sensitivity to heat and cold exposure remained normal [26]. Additionally, no difference was observed in the epidermal expression of other ion channels (TRPV1, TRPV3, TRPC1, TRPC3, Piezo 1, Piezo 2) in the mechanotransduction pathway [26]. Further investigation revealed that ATP release from glabrous skin in response to mechanical stimulation (20 mN for 10 s) and chemical stimulation (cinnamaldehyde) was significantly reduced in TRPA1-deficient mice, compared with wild-type mice [26]. Those findings support the essential role of TRPA1 in mechanosensation and signal transmission in murine keratinocytes. The impairment of mechanosensitivity observed in mice lacking TRPA1 emphasizes the importance of this channel in detecting and responding to mechanical cues in the skin.

In summary, these findings suggest that TRPA1 activation in keratinocytes plays a role in regulating processes related to skin barrier function, proliferation, differentiation, and inflammation. Understanding the mechanisms through which TRPA1 influences these processes can provide insight into potential therapeutic approaches for skin conditions that involve barrier dysfunction or inflammation.

#### 2.2.2. TRPV1

TRPV1 is a non-selective cation channel found in sensory neurons. It is known for its role in detecting and responding to a range of stimuli, including heat, protons (acidic conditions), and various environmental chemicals [58,59,60]. A recent focus in TRPV1 channel research is its activation by endocannabinoids, which has led to increased understanding of its role in various physiological phenomena [61]. Among endocannabinoids, anandamide (*N*-arachidonoylethanolamine, AEA) plays a crucial role in regulating the growth and survival of epidermal keratinocytes. Studies of cultured human keratinocytes and skin organ-culture models have reported that AEA inhibits keratinocyte proliferation and induces cell death, effects that are suppressed by TRPV1 inhibitors [62].

Notably, increased expression of TRPV1 was observed in the skin of psoriatic patients and in a mouse model of IMQ-induced psoriasis [14]. The overexpression of TRPV1 in pruritic skin correlates positively with the intensity of psoriasis itching. Indeed, TRPV1 knockout mice showed improvements in their inflammation levels and skin conditions [14]. Although most TRPV1 studies have focused on the cell membrane, the role played by TRPV1 in the endoplasmic reticulum (ER) membrane during inflammation has also been documented. Psoriatic skin has an observed thickening in the upper spinous layer of the epidermis, accompanied by increased expression of IL-1β and PAR2 [63]. Calcium depletion is thought to be the mechanism by which PAR2 activation causes inflammation, and TRPV1 in the ER plays an important role in that process. Both the inositol triphosphate receptor and TRPV1 in the internal calcium reservoirs play important roles in PAR2-evoked calcium release and subsequent skin inflammation in psoriasis [63].

#### 2.2.3. TRPV3

TRPV3 was initially identified as a temperature-sensitive cation channel that displays thermosensitivity within the range of innocuous warm temperatures (22 °C to 40 °C) [64]. TRPV3 is prominently expressed in epidermal keratinocytes, and its normal signaling is crucial for maintaining the homeostasis of the epidermal barrier. Recent studies have documented a significant increase in TRPV3 channel expression in severe scalp psoriatic lesions characterized by intense itching [13,65]. Importantly, gain-of-function mutations in TRPV3, identified in both mice and humans, are characterized by severe itching, hyperkeratosis, and elevated total IgE levels [29,66].

Molecular and cellular studies have investigated how TRPV3 causes itching and is involved in skin proliferation. As a result, TRPV3 is thought to be involved in the PAR2/Ca^2+^ signaling pathway within keratinocytes. It has been implicated in the development of chronic itch, and its inhibition has demonstrated efficacy in attenuating chronic itch models [67]. Moreover, EGFR-dependent signaling pathways are associated with TRPV3 and contribute to the promotion of skin keratinocyte proliferation. Activation of TRPV3 leads to a calcium influx that subsequently activates EGFR, PI3K, and NF-κB, ultimately resulting in cell proliferation [68].

#### 2.2.4. TRPV4

The TRPV4 channel is a non-selective cation channel that is activated by physiological temperatures, mechanical stimuli, and several endogenous mediators, including specific lipids, cholesterol, and their metabolic products [69,70,71]. Psoriasis involves a complex imbalance between cellular proliferation and differentiation. This pathological molecular signature is characterized by reduced levels of differentiation markers and increased expression of epidermal proliferation markers [72]. Notably, key differentiation markers include keratin 1 and keratin 10, which signifies their significant roles in the differentiation process [73]. During the process of cornification, the elevation of intracellular calcium plays a vital role in activating transglutaminase, enabling the crosslinking of structural proteins such as envoplakin and periplakin into keratin filaments [74].

TRPV4 plays a crucial role in selectively enhancing keratin production. Baicalein, the major flavonoid found in Scutellaria baicalensis root, inhibits the growth of keratinocytes and induces Ca^2+^-dependent differentiation through the activation of TRPV4 [12], which subsequently leads to ERK phosphorylation and increased expression of keratin 1 and keratin 10.

A recent study has shown that TRPV4 is upregulated in both human and mouse psoriasis models. In mouse models of chemically induced psoriasis, Trpv4-knockout mice exhibited milder dermatitis than wild-type mice. TRPV4 was found to be essential for mediating the expression of adenosine triphosphate derived from keratinocytes to activate and amplify immune responses [31]. Moreover, the expression levels of TRPV4 in the skin of three patients correlated positively with the severity of their psoriasis, and psoriasis-induced itch is strongly associated with TRPV4 activation [31].

A subcutaneous injection of GSK1016790A, a specific TRPV4 agonist, triggers acute itching in mice. Cimifugin is a compound that has traditionally been used in oriental medicine and effectively inhibits TRPV4 [75]. Notably, treatment with cimifugin mitigates the GSK1016790A-induced calcium response in HaCaT cells and dorsal root ganglion (DRG) neurons, thereby alleviating TRPV4-mediated acute and chronic itching [75]. The finding that TRPV4 expression correlates positively with the severity of psoriasis and itch suggests that pharmacological modulation of TRPV4 could be a promising therapeutic approach for managing psoriasis-induced itching.

#### 2.2.5. TRPC

The TRPC family, which includes seven mammalian channels, has been proposed to play roles as both store-operated and second messenger–operated channels in various cell types, including keratinocytes. In the context of keratinocytes, TRPC channels can be broadly categorized into two groups based on the mechanism of calcium influx. TRPC1 and TRPC4 channels are thought to play a role in store-operated calcium entry (SOCE), and TRPC6 and TRPC7 channels are associated with receptor-operated calcium influx. These distinctions reflect their specific functions in calcium regulation and signaling pathways within keratinocytes.

Maintaining calcium homeostasis through the activity of various TRPC channels is crucial for the normal functioning of the skin. TRPC7 is involved in mediating calcium entry in response to the presence of ATP and ATP/carbachol in keratinocytes [76]. Additionally, calcium entry mediated by TRPC1 and TRPC4 is notably enhanced in differentiated keratinocytes [77]. Moreover, the activation of TRPC6 has an inhibitory effect on keratinocyte proliferation. A compound called hyperforin, which acts as a TRPC6 agonist, effectively reduces keratinocyte proliferation [78].

In psoriatic keratinocytes, intracellular calcium levels following an increase in extracellular calcium are significantly lower than in healthy controls [35]. Psoriatic lesional skin shows a significant reduction in the expression of TRPC1, TRPC3, TRPC4, TRPC5, TRPC6, and TRPC7 channels, as well as the calcium-sensing receptor. This downregulation of TRPC channels could contribute to the disrupted calcium homeostasis observed in psoriatic keratinocytes [35]. Therefore, the regulation of calcium influx through TRPC channels is a critical component of calcium homeostasis within the skin, making it essential for maintaining proper skin function.

#### 2.2.6. TRPM4

TRPM4 is a monovalent cation channel that is activated by calcium ions and can be found in various cell types. Gain-of-function mutations in TRPM4 are the primary causative factor in erythrokeratodermia, a group of keratinization disorders characterized by erythematous (reddened), hyperkeratotic (thickened), sharply demarcated plaques [33,79].

Remarkably, TRPM4 mutations such as p.Ile1033Met and p.Ile1040Thr result in pronounced baseline activity, leading to elevated membrane potential and heightened calcium sensitivity that eventually causes hyperkeratotic skin [33]. In HaCaT cells, overexpression of the TRPM4 mutant results in increased cell proliferation compared with overexpression of wild-type TRPM4 [33]. Recent research using TRPM4 mutant mice has provided further evidence. In TRPM4 mutant (I1029M) transgenic mice (TRPM4I1029M), keratinocytes exhibited enhanced proliferation, and dendritic cells displayed enhanced migration [34]. Treatment with a TRPM4 inhibitor, glibenclamide, ameliorated the psoriasis induced by imiquimod in both wild-type and TRPM4I1029M mice, highlighting the potential therapeutic significance of TRPM4 targeting in the context of skin disorders.

### 2.3. CaSR and STIM/ORAI Channels

Cellular calcium signaling is a key regulator of impaired keratinocyte differentiation in psoriasis. The differentiation and proliferation of both murine and human keratinocytes are triggered by the extracellular calcium gradient. Notably, Menon and Elias were among the first to describe a defect in the epidermal calcium gradient in psoriatic skin [80,81,82].

Since then, several studies have demonstrated that the calcium homeostasis of keratinocytes in plaques from psoriatic patients is impaired by a substantial decrease in intracellular calcium influx/entry, compared with normal primary keratinocytes [35,36,83]. According to Karvonen et al., the function of store-operated calcium channels and capacitive calcium entry is significantly disturbed in psoriatic keratinocytes, despite the fact that psoriatic keratinocytes have slightly lower calcium influx stores than normal keratinocytes [83]. This abnormality in calcium levels could be attributable to disorders of calcium-regulating proteins. In this regard, key receptors and ion channels are involved, such as the calcium-sensing receptor (CaSR) responsible for sensing extracellular calcium ion levels and the stromal interaction molecule (STIM)/calcium release-activated calcium modulator (ORAI) complex that regulates SOCE [36,84,85].

Generally, keratinocyte differentiation is initiated by a rise in intracellular calcium levels in response to a high extracellular calcium concentration [35,83,86]. In this context, CaSRs play a pivotal role in regulating both the release of calcium from intracellular stores and the increase in calcium influx through calcium channels. Inhibition of CaSR expression induces a decline in intracellular calcium levels by increasing calcium reuptake into internal calcium stores and reducing Ca^2+^ influx via SOCE. As a result, this inhibition impedes the differentiation of keratinocytes by disabling their ability to respond to extracellular calcium with changes in intracellular calcium levels. Thus, CaSR-mediated calcium signaling plays a crucial role in maintaining intracellular calcium homeostasis and promoting the differentiation of keratinocytes [87,88,89]. SOCE is an essential pathway that is activated when calcium stores in the ER become depleted. It depends on two primary components, STIM1, situated within the ER and acting as a calcium sensor, and ORAI1, which functions as a subunit of calcium channels. The interaction between STIM1 and ORAI1 is a fundamental process in cellular calcium signaling and plays essential roles in controlling the differentiation and proliferation of keratinocytes [36,90,91]. In response to the depletion of ER calcium stores, STIM orchestrates localized activation of calcium influx to replenish the ER through the selective co-localization of STIM with ORAI1 channels at the ER and plasma membrane. Thus, their dynamic coordination maintains the cellular calcium balance and establishes the fundamental unit of SOCE [90,91,92].

Cubillos et al. demonstrated that the CaSR, STIM1, and ORAI channels (ORAI1, ORAI3) are diminished in the plaques of patients with psoriasis vulgaris [36]. Reducing the expression of either STIM1 or ORAI1 leads to alterations in intracellular calcium stores, SOCE inhibition, and the abolishment of calcium responses to changes in extracellular calcium levels, which hinders the differentiation and proliferation of undifferentiated keratinocytes [90]. Therefore, these calcium-modulating proteins play a crucial role in maintaining intracellular calcium balance by detecting extracellular Ca^2+^ concentration changes and regulating the release of Ca^2+^ from intracellular stores, which, in turn, affects the differentiation and proliferation of keratinocytes [87].

### 2.4. Chloride Channels

Chloride ion channels are transmembrane proteins responsible for controlling the movement of chloride ions (Cl^−^) across the cell membrane. These channels serve as vital components in various physiological functions, such as muscle contraction, fluid secretion, and neuronal excitability [93].

Interestingly, recent studies have suggested that chloride channels could play a role in the pathogenesis of psoriasis. ANO1 is a calcium-activated Cl^−^ channel that participates in regulating transmembrane fluid secretion in the epithelial cells of various organs [94,95,96,97,98]. In sensory neurons, ANO1 has also been implicated in nociceptive (pain) and pruritic signaling pathways [95,99]. Notably, recent research has highlighted the significant expression of ANO1 in keratinocytes. In this context, ANO1 exerts regulatory control over the ERK and AKT signaling pathways [9]. Imiquimod is a topical medication used in dermatology for various purposes [100,101]. A remarkable study demonstrated that the application of IMQ induced the hyperproliferation of keratinocytes, leading to increased ANO1 expression and thicker epidermal layers [9]. The application of an ANO1 inhibitor (T16Ainh-A01) inhibited the IMQ-induced keratinocyte hyperproliferation and suppressed the production of inflammatory mediators such as interleukins and TNF-α [9]. The activation of ANO1, dependent on intracellular calcium increases, underscores the importance of understanding its interactions with TRP channels. For instance, ANO1-mediated currents have been observed in response to the activation of TRPV3 in epidermal keratinocytes [28]. The coupling mechanism between ANO1 and TRPV3 has a modulatory effect on wound healing processes, primarily by regulating cell proliferation and migration through the modulation of p38 phosphorylation [28].

Similar to ANO1, the cystic fibrosis transmembrane conductance regulator (CFTR) is a chloride ion transporter that plays a role in regulating fluid and ion balance across various epithelial tissues, including keratinocytes. CFTR-deficient mice (ΔF508 cftr^−/−^ mice) exhibited delayed skin recovery in a dorsal skin wounding model that damaged both the epidermis and the underlying dermis, compared with wild-type mice [102]. Thereafter, Chen et al. found that CFTR helps alleviate inflammation, reduce proliferation, and promote differentiation in keratinocytes by suppressing MAPK/NF-κB signaling, which ultimately facilitates cutaneous wound healing [103].

LRRC8a is a component of volume-regulated anion channels that regulate the hypotonic stress response of human keratinocytes [104]. This function is of particular significance because environmental osmotic fluctuations constantly challenge the barrier function of the human epidermis. A recent study revealed that LRRC8a regulates the cell proliferation of human metastatic cells by interacting with ANO1 [105]. Taken together, these findings imply that chloride channels could be a potential avenue for therapeutic interventions to treat psoriasis.

### 2.5. Voltage-Gated Sodium Channels

Voltage-gated sodium (Nav) channels play crucial roles in the electrical signaling of various cell types. Nav channels are a family of nine distinct subtypes, with Nav1.1, Nav1.2, and Nav1.3 found in the central nervous system and Nav1.6 distributed in both the peripheral and central nervous systems. In contrast, Nav1.7, Nav1.8, and Nav1.9 are predominantly localized in the peripheral nervous system [106], and Nav1.4 and Nav1.5 channels are abundant in skeletal and cardiac muscles, respectively.

Nav channels are primarily known for their critical roles in neurons and muscle cells, particularly in the initiation and propagation of action potentials. However, several reports suggested that they also have a role in regulating the function of keratinocytes. Nav1.1, Nav1.6, and Nav1.8 have been detected in keratinocytes within the rat epidermis [107]. Although Nav1.8 mediates nociceptive and inflammatory signaling in sensory neurons, a remarkable study showed that increased levels of Nav1.8 in the skins epidermal layer could contribute to heightened pain sensitivity [108]. Additionally, Zhang et al. demonstrated a significant upregulation of Nav1.8 in the epidermis of psoriatic skin, which is strongly associated with heightened pain sensitivity [37]. A further mechanistic study elucidated how Nav1.8 might contribute to the inflammatory processes in psoriatic skin [37]. The cytokine TNF-α exacerbates various inflammatory skin conditions, and treatment with TNF-α induces an upregulation of Nav1.8 expression. The increased expression and activity of Nav1.8, in turn, altered the redox balance within the affected skin tissue. In particular, the interaction between Nav1.8 and SOD2, a critical antioxidant enzyme responsible for neutralizing superoxide radicals, led to a reduction in SOD2 activity that could contribute to ROS accumulation. Thus, Nav could play roles in inflammatory processes in the skin and potentially exacerbate skin diseases such as psoriasis.

### 2.6. Mechanosensitive Channels

Mechanotransduction is a crucial process by which mechanical forces are converted into electrical or biochemical signals in cells. Such signals play pivotal roles in maintaining cellular homeostasis, influencing various aspects of keratinocytes, including their proliferation and differentiation. The significance of mechanical stress in triggering psoriasis is notable, as exemplified by the Koebner phenomenon [109]. This phenomenon involves the development of psoriatic plaques on previously healthy skin after exposure to trauma or mechanical stressors such as scratches, abrasions, and pressure. Ion channels play a significant role in mechanotransduction because they allow ions to flow in response to mechanical forces. Some key ion channels are piezo, polycystin, and TRP channels [110].

The PIEZO1 mechanosensitive channel is functionally expressed in both mouse and human keratinocytes, where it plays a crucial role in initiating sensory afferent firing in response to mechanical stimuli [111]. Thus, PIEZO1 plays a role in the perception of mechanical stimuli in the skin, which is a crucial aspect of sensory function. However, the epidermis of PIEZO1-conditional knockout mice exhibited normal morphological features, despite the functional expression of PIEZO1 in keratinocytes [111], suggesting that PIEZO1 might not be indispensable to the basic structural development of the epidermis. Therefore, considering the functional expression of PIEZO1 in sensory processes and skin function is not only of scientific interest but also holds potential implications for conditions involving sensory perception and mechanosensory deficits. Further research is needed to fully comprehend the complexities of mechanosensation in the skin.

Polycystin (PC) has emerged as a mechanosensor in epithelial cells. PC1 is a transmembrane protein with a mechanosensitive *N*-extracellular end and a transcriptionally active cleavage-inducing C-terminus, and PC2 is a non-selective cation channel belonging to the TRP family of channels, specifically as a TRPP channel. PC1 is expressed in keratinocytes, where it is downregulated in psoriatic lesions [38]. The downregulation of PC1 could contribute to the development of psoriatic plaques, potentially through the mTOR pathway. Activation of mTOR and its downstream substrates has been observed in psoriatic lesions, and inhibiting mTOR has shown promise in reducing keratinocyte proliferation [112]. The signaling pathways of ion channels in keratinocyte are summarized in Figure 2.

## 3. Ion Channels in Immune Cells

### 3.1. TRP Channels

Some TRP channels are expressed in various types of immune cells, including T cells [113]. Various studies have shown that TRP channels have distinct effects on the dermal immune system. For instance, TRPA1 expression was detected in dermal CD4+ T cells in psoriasiform skin inflammation induced by IMQ. Indeed, TRPA1-overexpressing cells are directly activated by IMQ, and that effect is attenuated in the presence of HC030031, a specific TRPA1 inhibitor [17]. IMQ-induced psoriasiform skin inflammation was exacerbated in TRPA1-deficient mice. Additionally, there was a more pronounced increase in T helper-associated cytokines in the TRPA1-deficient mice, suggesting that TRPA1 plays a potential role in regulating CD4+ T cell signaling.

On the other hand, the TRPV1 channel expressed in CD4+ T cells contributes to T cell receptor–induced calcium influx, thereby enhancing their pro-inflammatory characteristics [16]. Consistently, the area of Munro micro-abscesses and psoriasis-associated angiogenesis was significantly reduced in TRPV1-deficient mice [14]. IMQ-induced psoriatic lesions in TRPV1-deficient mice showed lower mRNA levels of pro-inflammatory cytokines (IL-1β, IL-6, IL-23, S100A8, and CXCL1) and higher mRNA levels of the immunosuppressive cytokine IL-10, compared with their wild-type counterparts [114], suggesting that TRPV1 increases the expression of pro-inflammatory cytokines and reduces the expression of immunosuppressive cytokines in psoriasis, thereby contributing to the development of psoriatic inflammation. Therefore, the contrasting functions of TRPA1 and TRPV1 in CD4+ T cells collectively contribute to the regulation of CD4+ T cell activity in mediating inflammation [115].

Recent studies have examined the gene expression profiles of human psoriasis patients. Those investigations have revealed changes in the gene expression of TRP channels within immune cells. Specifically, peripheral blood mononuclear cells from individuals with psoriasis exhibit elevated expression of TRP channels such as TRPM2 and TRPV1 and decreased expression of TRPM4, TRPM7, TRPV3, TRPV4, and TRPC6 [27]. These findings underscore the complex involvement of TRP channels in the immune responses and inflammation associated with psoriasis. Understanding how changes in the expression of these channels influence immune cell behavior and skin inflammation is a promising avenue for future research.

### 3.2. STIM/ORAI in Immune Cells

Psoriasis stands out among other inflammatory skin conditions due to its complicated involvement of various inflammatory cells, including macrophages, lymphocytes, and neutrophils. The interplay between those immune cells and keratinocytes, which results in the production of cytokines, chemokines, and growth factors, is thought to drive the disease [116].

Several studies have underscored the importance of calcium signaling mediated by STIM1/ORAI1 in neutrophil chemotaxis. Chemo-attractants such as CXCL-1, macrophage inflammatory protein-2, and IL-8 bind to CXC receptors, initiating phospholipase C activation and IP_3_ generation. The produced IP_3_ binds to IP_3_ receptors in the ER, leading to the release of stored calcium. Depletion of STIM1 in neutrophils impedes their ability to infiltrate IMQ-induced psoriatic skin lesions, and STIM1 knockout mice exhibit a more rapid recovery from the psoriasis-like lesions induced by IMQ than their wild-type counterparts [117].

Given the immunomodulatory potential associated with calcium release–activated channels (CRAC), ongoing research is targeting those channels for immune system regulation. Celastrol, a natural compound, has emerged as a novel SOCE inhibitor and shown efficacy in ameliorating skin lesions and reducing psoriasis symptoms in a murine IMQ-induced psoriasis model [118]. Furthermore, recent studies have introduced two novel and highly selective CRAC channel inhibitors: C63368, which has an indole-like structure, and C79413, which contains a pyrazole core. These inhibitors have demonstrated potent and reversible effects in inhibiting the CRAC channel at low concentrations [119]. Both compounds effectively suppressed Jurkat cell proliferation and cytokine production in human T lymphocytes. When administered intragastrically to mice, these inhibitors delivered significant therapeutic benefits in psoriasis models [119].

In addition to CRAC channels, the arachidonic acid–regulated calcium-selective (ARC) channel is also implicated in psoriasis. ARC channels have biophysical properties very similar to CRAC channels, with the key distinction that the CRAC channel consists of the Orai1 subunit, whereas the ARC channel is formed by the Orai1 and Orai3 subunits [18,19]. Lesional psoriatic skin is characterized by an excess of lipid and fatty-acid mediators, including metabolites derived from both arachidonic and linoleic acids [120]. Moreover, patients with psoriatic arthritis display increased expression of ORAI3 in CD4+ T cells, leading to enhanced ARC channel activity and heightened sensitivity to arachidonic acid [20]. The reduced expression of IKAROS, a transcriptional repressor of the ORAI3 promoter, is responsible for the elevated ORAI3 transcription in T cells, which in turn contribute to enhanced ARC channel activity. Arachidonic acid not only induces calcium influx in T cells through ARC activation but also triggers the phosphorylation of components in the T cell receptor signaling cascade [20].

### 3.3. nAChR

The α7 receptor, a homo-oligomeric nAChR, is not only present in the central nervous system but is also expressed in non-neuronal cells, including keratinocytes. In non-neuronal contexts, α7 receptors play pivotal roles in processes such as angiogenesis, cell-cycle progression, and metastasis [121]. In addition, α7 receptors are found in immune system cells such as lymphocytes, monocytes, and macrophages [122]. In macrophages, acetylcholine-mediated α7 receptor activation initiates the cholinergic anti-inflammatory pathway [122], suppressing the release of TNF and other cytokines through a post-transcriptional mechanism [123]. This anti-inflammatory pathway triggered by the stimulation of α7nAChR operates through the JAK2–STAT3 signaling pathway [124].

In mouse skin, α7nAChR has been identified in monocytes and fibroblasts, as well as implicated in the skin wound healing process [125]. The role of nAChRs has also been explored in human psoriasis patients. Notably, guttate psoriasis lesions are characterized by degranulated mast cells, which can be activated by various stimuli, including acetylcholine [126]. In these psoriasis lesions, both acetylcholine production and AChR expression tend to shift from the basal to the supra-basal layers of the epidermis [126]. In vitro experiments have shown that the stimulation of nicotinic receptors can induce the complete degranulation of human mast cells [126].

Furthermore, research into acetylcholine receptors in the immune cells of patients with rheumatoid arthritis and psoriatic arthritis has revealed the significant expression of α7nAChR. Notably, psoriatic arthritis patients exhibited higher total α7nAChR expression than those with rheumatoid arthritis [23]. Acetylcholine and an α7nAChR agonist reduce interleukin production, an effect that can be blocked by an α7nAChR antagonist [22]. Consequently, modulation of 7nAChR holds promise as a potential therapeutic target for psoriasis.

### 3.4. Potassium Channels in Psoriasis

Voltage-gated potassium (Kv) channels play a critical role in regulating the proliferation, differentiation, and apoptosis of effector memory T cells. Recent research has emphasized the significance of Kv1.3 channels in the development of psoriasis, particularly in controlling T cell activation and proliferation.

In psoriatic dermis and synovial tissue from individuals with psoriatic arthritis, Kv1.3 channels are notably overexpressed in infiltrating T cells [40]. This high expression of Kv1.3 channels was also observed in psoriatic lesions from psoriasis patients and in human psoriasiform skin grafts onto a severe combined immunodeficient mouse model [39]. Remarkably, the administration of Stichodactyla helianthus neurotoxin (ShK), a well-known Kv1.3 blocker, demonstrated significant therapeutic benefits in psoriasiform skin grafts, suggesting Kv1.3 blockers as potentially novel therapeutic agents for psoriasis [39]. ShK-186, a synthetic analog of ShK, has undergone IND-enabling toxicity studies and recently completed human phase 1A and phase 1B trials in healthy volunteers. Another Kv1.3 blocker, PAP-1, dose-dependently inhibited proliferation and suppressed IL-2 and IFN-γ production in in vitro studies performed with lesional mononuclear cells or T cells derived from the skin and joints of psoriatic patients [127]. To evaluate the efficacy of PAP-1 on the skin, it was mixed with skincare cream such as Eucerin for topical application [40]. Mixing Kv blockers with cream formulations for skin conditions can potentially yield synergistic effects in psoriasis treatment. As Kv1.3 is also expressed in neurons of the olfactory bulb and auditory system, Kv1.3 blockers should be used with caution due to their potential impact on smell or high-frequency hearing [128,129]. Therefore, this topical application method is more convenient and can be expected to have fewer side effects compared to systemic injection or intradermal injections.

Notably, Lozano-Gerona et al. explored how overexpression of the cell volume–regulating Ca^2+^-activated KCa3.1 channel affected murine epidermal homeostasis. Mice treated with doxycycline and carrying the KCa3.1+ transgene under the control of the reverse tetracycline-sensitive transactivator showed 800-fold channel overexpression above basal levels in the skin, leading to epidermal spongiosis, progressive epidermal hyperplasia and hyperkeratosis, itch, and ulcers. Those effects were accompanied by the production of pro-proliferative and pro-inflammatory cytokines (IL-1β, IL-6, and TNF-α) in the skin. Treatment with the KCa3.1-selective blocker Senicapoc significantly suppressed spongiosis and hyperplasia, as well as the induction of IL-1β and IL-6. That study thus identified KCa3.1 as a regulator of epidermal homeostasis and spongiosis and a potential therapeutic target for eczematous dermatitis [21]. The signaling pathways of ion channels in immune cells are summarized in Figure 3.

## 4. Ion Channels in Peripheral Sensory Neurons

Psoriasis is primarily considered a skin condition, but it can have effects beyond the skin, including interactions with the peripheral nervous system. In psoriatic pathology, ion channels in peripheral sensory neurons mediate itch and pain sensation and regulate the immune system.

### 4.1. Peripheral Itch Pathway

A typical symptom of psoriasis is pruritus. Among peripheral sensory neurons, pruritus is mediated by unmyelinated C-fibers and myelinated A delta-fibers. These fibers convey pruritic signals from their distal termini to the neuronal cell bodies in the DRG or trigeminal ganglion. Nociceptive neurons, including itch-sensitive neurons, are excitatory glutamatergic neurons that express vesicular glutamate transporter type 2 (VGLUT2) [130,131]. Genetic knockout of VGLUT2 in all nociceptive neurons resulted in the abolishment of pain behaviors but enhanced itch [131]. Thus, itch-mediating sensory neurons do not exclusively rely on glutamate for synaptic transmission but instead have been proposed to communicate through neuropeptides.

The nerve growth factor (NGF) neuropeptide, which acts through a TRPV1 pathway, may be a crucial regulator of neurogenic pruritus [24]. NGF is released at peripheral nerve endings and forms a complex when binding to the TrkA receptor that leads to the sensitization of primary afferent nociceptors to thermal and chemical stimuli in vitro and in vivo [132]. The NGF signaling cascades in primary afferent neurons result in sensitization or increased expression of several receptors and channels on the membrane surface. These include TRPV1, acid-sensing ion channels, voltage-gated sodium and calcium channels, delayed rectifier potassium currents, and putative mechanotransducers. These changes contribute to immediate hypersensitivity following inflammation [133,134].

Moreover, the gastrin-releasing peptide receptor (GRPR) assumes a pivotal role in mediating the sensation of itch in the dorsal spinal cord [135]. Notably, gastrin-releasing peptide exhibits specific expression within a small subset of peptidergic DRG neurons, and GRPR expression is confined to the lamina I of the dorsal spinal cord [135]. The GRPR-positive neurons in the lamina I of the spinal cord are excitatory interneurons that depend on the VGLUT2 for neurotransmission [136]. Additionally, a population of neuropeptide Y–positive inhibitory interneurons within the spinal cord suppresses mechanical itch [137]. Moreover, natriuretic polypeptide B (NPPB, also referred to as B-type natriuretic peptide, BNP), which is expressed in TRPV1-positive neurons, is strongly associated with itch. NPPB-genetic depletion demonstrated a reduction in the scratching behavior induced by histamine, chloroquine, 5-HT, and the SLIGRL peptide [138].

Although TRPV1 is the primary ion channel involved in the transmission of itch signals in nociceptors, it is noteworthy that other ion channels in the DRG have also been implicated in pruritus induction by psoriasis [8,139]. Human psoriasis lesions exhibit elevated expression of several microRNAs, including miR-203b-3p. Recent research has revealed that miR-203b-3p induces calcium ion influx within the DRG, inciting scratching behavior in mice through the activation of 5-HTR2B and the PKC-dependent phosphorylation of TRPV4 [25]. TRPC4 has also been reported to contribute to scratching behavior by inducing calcium influx within the downstream pathway of peripheral serotonin receptors [5]. Additionally, ANO1 expressed in DRG has been identified as a participant in itch transmission, particularly in instances induced by Mrgpr activation [99].

### 4.2. Immune System Regulation

Most mast cells are localized in the papillary dermis, specifically at the dermal–epidermal junction, within pruritic skin. In psoriatic conditions, mast cells undergo degranulation, leading to the release of granules that contain various pruritus-inducing molecules such as cytokines, histamine, and serotonin. The released substances subsequently interact with specific receptors on axon terminals, initiating the activation of the TRPV1 channels present on sensory nerve fibers [24]. NGF is also a potent degranulation factor for mast cells [140]. The NGF-mediated TRPV1 signaling pathway serves as a pivotal regulator of neurogenic pruritus, concurrently contributing to the establishment of a dynamic neuroimmune network. Substance P (SP), a highly conserved sensory peptide, exhibits a strong and intricate correlation with the NGF-TRPV1 axis, augmenting the complexity of this neuroimmune network. NGF stimulates SP synthesis and expression. When SP is then released, it reciprocally induces the synthesis of NGF through the involvement of inflammatory mediators. In addition, binding between SP and the neurokinin-1 receptor potentiates the opening of TRPV1, which can also promote SP release [141,142].

In the context of psoriasis, the IL-23/IL-17 axis, which is a pathogenic pathway, is significantly potentiated by keratinocytes in an IMQ-treated model [143,144]. This activation of the immune system is intimately linked to peripheral nociceptive neurons. More specifically, a distinct subset of sensory neurons expressing TRPV1 and Nav1.8 ion channels plays an indispensable role in driving this inflammatory response [145]. Imaging studies of intact skin have revealed that dendritic cells, the primary source of IL-23, interact closely with these nociceptors. Intriguingly, genetic ablation of Nav1.8/TRPV1-positive nociceptors results in dendritic cells that fail to generate IL-23 in skin exposed to IMQ [145]. Additionally, Zhou et al. reported a substantial reduction in epidermal hyperplasia, inflammatory cell infiltration, and cytokine production in IMQ-treated TRPV1 knockout mice [14]. More recent investigations have shed light on the intricate mechanisms by which nociceptive sensory neurons can stimulate the production of IL-23 and IL-17 by γδ T cells through the action of the neuropeptide CGRP, thus contributing to the pathology of psoriasis [146,147].

### 4.3. Expression Change in Neuronal Cells in Psoriasis

Sakai et al. reported that the expression patterns of various TRP channels change in sensory neurons affected by psoriasis. Despite the observed increase in TRPV1 expression in psoriatic keratinocytes, a mouse model of psoriatic dermatitis did not exhibit significant changes in TRPV1 and TRPA1 mRNA expression levels in DRG cells. Interestingly, the researchers noted that TRPV4 mRNA expression exhibited a transient increase in DRG cells, whereas TRPM8 mRNA levels were significantly reduced [32]. These findings suggest that TRP channels other than TRPV1 might have a close association with the inflammatory mechanisms underlying psoriasis.

### 4.4. Therapeutic Targets

TRPV1 has long been studied as a clinical target for psoriasis treatment. Given that the initial vasodilation and leakiness of papillary vessels may be crucial events in psoriasis development, agents capable of blocking these vascular reactions could be effective in treating the disorder.

Capsaicin has been shown to effectively inhibit cutaneous vasodilatation induced by various chemical and physical stimuli when applied topically. Capsaicin stimulates TRPV1, causing neurons to release substance P. However, continuous application of capsaicin has the effect of blocking TRPV1-expressing neurons. By depleting substance P at local sensory nerve terminals, capsaicin can suppress vasodilation and leakiness of papillary vessels, thereby inhibiting the early development of psoriasis [148].

Furthermore, recent research has revealed that percutaneous capsaicin delivery effectively hindered the activation of the IL-23/IL-17 pathway, resulting in a significant reduction in psoriasiform dermatitis. The application of capsaicin led to a marked reduction in the cytokines induced by IMQ [149].

Chan et al. suggested that although the activation of TRPV1 in non-neuronal cells can exhibit pro-inflammatory properties, the blockade of TRPV1 on nerve cells might exert a more potent inhibitory influence, particularly during the early stages of psoriasiform inflammation [149].

Capsaicin treatment can stimulate TRPV1 at nerve endings in the skin, leading to sensations of burning, stinging, itching, and redness. However, it has been reported that these sensations mostly diminish or vanish with continued application [148].

Xanthotoxin, a naturally occurring furanocoumarin with established clinical applications for psoriasis treatment, exerts therapeutic effects by decreasing TRPV1 activity and expression in the DRG and inhibiting immune responses [150]. Additionally, resolvin D3 has shown potential in reducing acute TRPV1-dependent pain and itch in mice. Systemic administration of resolvin D3 prevents the development of both psoriasiform itch and skin inflammation, coinciding with a reduction in the expression of the neuropeptide CGRP in DRG neurons [151].

Additionally, a TRPC4 inhibitor, ML204, significantly inhibited psoriatic itch and cutaneous inflammation when administered via intradermal injections in IMQ-treated skin [152]. Conversely, it has been suggested that TRPM8 activators might serve as potential antipruritic compounds. Thymol, in particular, has shown the ability to reduce scratching behavior and alleviate the upregulated expression of pro-inflammatory cytokines in IMQ-treated mice [153]. The signaling pathways of ion channels in sensory neurons are summarized in Figure 4. These findings collectively underscore the potential of TRP channel–induced calcium influx as a therapeutic approach for managing pruritus and the associated inflammatory response in psoriasis.

## 5. Perspectives

This review has discussed the biological mechanisms of ion channels involved in psoriasis and their role in the dysregulation of epidermal cells, immune responses, and sensory neuron signaling. The dysregulation of ion channels in psoriasis plays a significant part in the pathophysiology of the condition, affecting various aspects of epidermal cells, immune responses, and sensory neuron signaling (Table 1). Dysregulation of these channels can lead to altered calcium signaling, inflammation, proliferation, and sensory signaling, all of which are central features of psoriasis. The interactions among the skin, immune system, and sensory nerves are multifaceted, and a deeper understanding of their connections could lead to more targeted and effective treatments for psoriasis-related symptoms such as itching and pain.

Several ion channels function as positive triggers for psoriasis. Overexpression, gain of function, and activation of channels, including nAChR, TRPV1, TRPV3, TRPV4, TRPM4, and ANO1 in keratinocytes, as well as TRPV1, nAChR, Kv1.3, and KCa3.1 in immune cells, have been associated with the induction of psoriasis. Thus, inhibitors directed at these channels could be potential candidates for therapeutic intervention. On the contrary, polycystin, CFTR, TRPC, and CRAC channels in keratinocytes exert repressive roles on psoriasis development. Hence, agonists targeting these ion channels in keratinocytes could be potential substances for psoriasis treatment.

Interestingly, the TRPA1 channel appears to have dual functions—both positive and negative—in relation to psoriasis pathogenesis, depending on the cell type expressing it. Activation of TRPA1 in keratinocytes is implicated in the development of psoriasis, while its activation in immune cells is speculated to have suppressive effects on psoriasis pathogenesis. Additionally, it is noteworthy that among nAChR types, alpha5 exhibits pathological effects on psoriasis, whereas alpha7 shows a negative impact (alpha7 agonists have been shown to alleviate psoriasis symptoms).

Therefore, selectively modulating specific ion channel types or subunits can indeed lead to more effective results (Table 2 and Table 3). Additionally, understanding the interaction between different cell types, such as keratinocytes, immune cells, and sensory neurons, is essential for developing targeted therapies. Research efforts should focus on identifying ways to target ion channels within these specific cell types to tailor treatments that address the intricate dynamics of psoriasis pathogenesis. This comprehensive approach could pave the way for more precise and efficient interventions in the management of psoriasis.

## Figures and Tables

**Figure 1 ijms-25-02756-f001:**
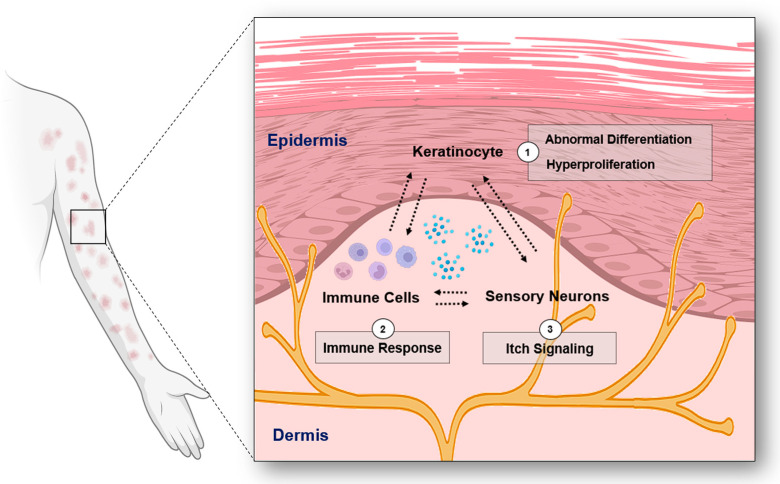
The overview of the interplay among keratinocytes, immune cells, and sensory neurons in psoriasis (Figure image was created with BioRender.com (accessed on 31 December 2023)).

**Figure 2 ijms-25-02756-f002:**
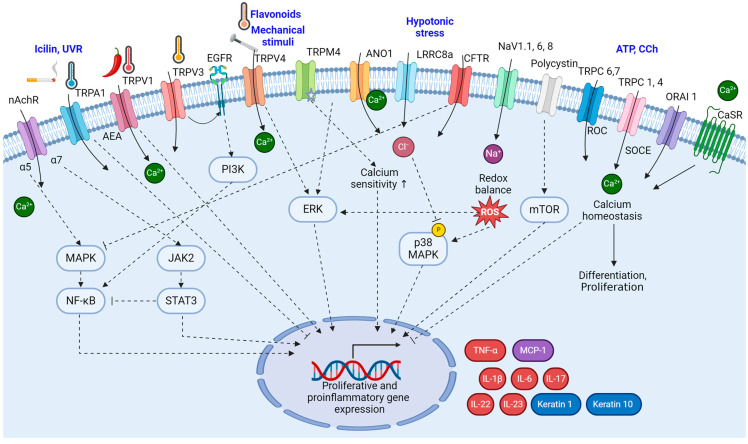
The signaling pathways of ion channels in psoriatic keratinocytes. The signaling pathways involving ion channels in psoriatic keratinocytes are interactive and contribute significantly to the altered cellular functions observed in psoriasis. The intricate interplay between these signaling pathways, influenced by ion channel activity, creates a feedback loop that perpetuates the altered cellular functions observed in psoriasis. (Figure image was created with BioRender.com (accessed on 22 November 2023)).

**Figure 3 ijms-25-02756-f003:**
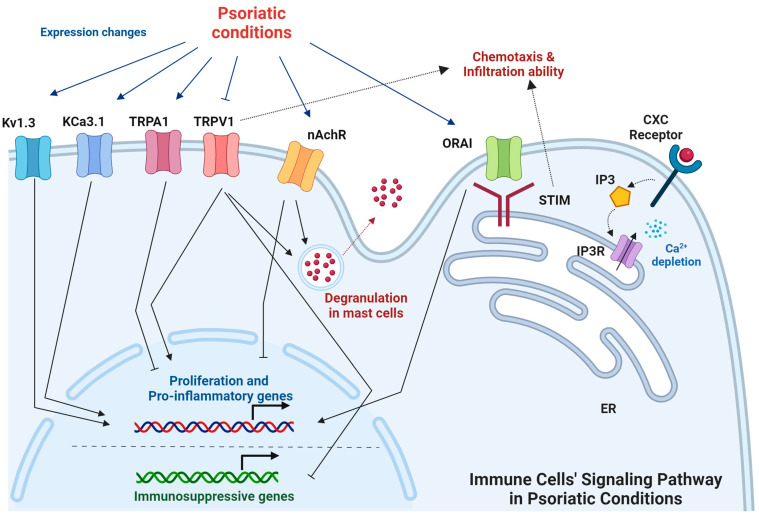
The signaling pathways of ion channels in immune cells under psoriatic conditions. The signaling pathways involving ion channels in immune cells under psoriatic conditions are interactive and regulate the characteristics of immune cells. (Figure image was created with BioRender.com (accessed on 7 February 2024)).

**Figure 4 ijms-25-02756-f004:**
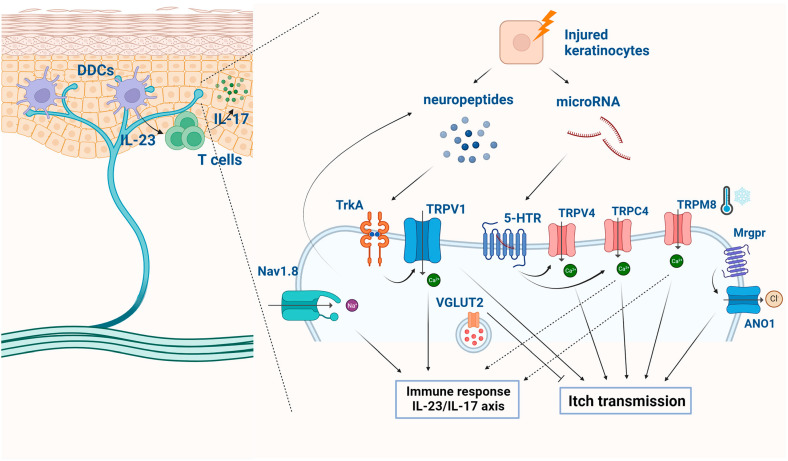
The signaling pathways of ion channels in sensory neurons in psoriasis. The signaling pathways involving ion channels in sensory neurons contribute significantly to the sensory symptoms, such as itch and pain. (Figure image was created with BioRender.com (accessed on 5 December 2023)).

**Table 1 ijms-25-02756-t001:** Ion channels related to psoriatic diseases. (↑: increased expression, ↓: decreased expression).

Type of Ion Channels	Expression Change in Psoriasis	Mutation
nAChR	α5 ↑ in keratinocytes [11] α7 ↑ in keratinocytes [4] ↑ in immune cells [23]	
TRPA1	↑ in keratinocytes [26]	
TRPV1	↑ in keratinocytes [14] no change in keratinocytes [26] ↑ in immune cells [27]	
TRPV3	↑ in keratinocytes [28] ↓ in immune cells [27]	R416Q, R416W, L655P, W692S, L694P, G568D, G568V, L673F: ion channel activity ↑ severe Olmsted syndrome (L673F, W692S) mild Olmsted syndrome variants (R416Q) [29] G573A [30]
TRPV4	↑ in keratinocytes [31] ↓ in immune cells [27] ↑ in neurons [32]	
TRPM	TRPM2 ↑ TRPM4, M7 ↓ in immune cells [27] TRPM8 ↓ in neurons [32]	I1033M, I1040T (human): ion channel activity ↑ [33] I1029M (mouse): ion channel activity ↑ [34]
TRPC	TRPC1,3,4,5,6,7 ↓ in keratinocytes [35] TRPC6 ↓ in immune cells [27]	
CaSR	↓ in epidermis [36]	
STIM/ORAI	↓ in epidermis [36]	
ANO1	↑ in keratinocytes [9]	
Nav	Nav1.8 ↑ in keratinocytes [37]	
Polycystin	PC1 ↓ in keratinocytes [38]	
Kv	Kv1.3 ↑ in epidermis [39,40]	
KCa	Kca3.1 ↑ in epidermis [21]	

**Table 2 ijms-25-02756-t002:** Drug candidates acting on ion channels for psoriasis treatment. (↑: increase, ↓: decrease).

Drugs	Target Ion Channel	Cell Type	Mechanisms	Functions	Tested Experimental Models
PNU-282987	nAChR	keratinocytes immune cells	nAChR activation	inflammation ↓ keratinocyte proliferation ↓ abnormal differentiation ↓	Animal study: i.p. injection
AR-R17779	nAChR	keratinocytes immune cells	nAChR activation	inflammation ↓	Animal study: i.p. injectionhuman primary epidermal keratinocytes
Tropisetron	nAChR	keratinocytes	nAChR activation	inflammation ↓ collagen synthesis ↓	human primary epidermal keratinocytes
Tacrolimus	TRPA1	keratinocytes	TRPA1 expression ↓	inflammation ↓	HaCaT cells
Cyclosporine	TRPA1	keratinocytes	TRPA1 expression ↓	inflammation ↓	HaCaT cells
Dexamethasone	TRPA1	keratinocytes	TRPA1 expression ↓	inflammation ↓	HaCaT cells
Resveratrol	TRPA1	keratinocytes	TRPA1 expression ↓ inflammatory ROS ↓	Inflammation ↓ keratinocyte differentiation ↓	HaCaT cells
Anandamide	TRPV1	keratinocytes	TRPV1 activation	keratinocyte proliferation ↓ cell death ↑	human cultured keratinocytes, skin organ-culture models
Xanthotoxin	TRPV1	keratinocytes	TRPV1 activity ↓ TRV1 expression	inflammation ↓ antinociceptive activity	Animal study: oral administration
Resolvin D3	TRPV1	peripheral neurons	TRPV1 activity ↓	itch ↓ skin inflammation ↓	Animal study: intradermal/systemic injection
Capsaicin	TRPV1	peripheral neurons	TRPV1 desensitization	IL-23/IL-17 pathway inhibition	Animal study: Topical treatment
ML204	TRPC4	peripheral neurons	TRPC4 inhibition	itch ↓ skin inflammation ↓	Animal study: Intradermal injection
Thymol	TRPM8	peripheral neurons	TRPM8 activation	itch ↓ skin inflammation ↓	Animal study: Subcutaneous injection
Erlotinib	TRPV3	keratinocytes	EGFR inhibition (TRPV3 downstream pathway)	keratinocyte proliferation ↓	HacaT cells
BAY11-7085	TRPV3	keratinocytes	NF-κB inhibition (TRPV3 downstream pathway)	keratinocyte proliferation ↓	HacaT cells
LY294002	TRPV3	keratinocytes	PI3K inhibition (TRPV3 downstream pathway)	keratinocyte proliferation ↓	Animal study: subcutaneous injection
Baicalein	TRPV4	keratinocytes	TRPV4 activation, K1/K10 expression ↑	keratinocyte proliferation ↓ keratinocyte differentiation ↑	HacaT cells
Cimifugin	TRPV4	peripheral neurons	TRPV4 inhibition in peripheral neurons	itch ↓	Animal study: subcutaneous injection
Glibenclamide	TRPM4	keratinocytes immune cells	TRPM4 inhibitor, DC migration ↓	inflammation ↓	Animal study: i.p. injection
T16Ainh-A01	ANO1	keratinocytes	ANO1 inhibition, ERK and AKT signaling ↓	keratinocyte proliferation ↓ inflammation ↓	Animal study: skin treatment
C63368	CRAC	immune cells	CRAC channel inhibitors	immune cell proliferation ↓ cytokine production ↓	in vitro: jurkat cells in vivo: skin treatment
C79413	CRAC	immune cells	CRAC channel inhibitors	immune cell proliferation ↓cytokine production ↓	in vitro: jurkat cells in vivo: skin treatment
Stichodactyla helianthus neurotoxin	Kv1.3	immune cells	Kv1.3 inhibitor	immune cell proliferation ↓ IL-2 and IFN-γ production ↓	Animal study: intradermal injection
PAP-1	Kv1.3	immune cells	Kv1.3 inhibitor	immune cell proliferation ↓ IL-2 and IFN-γ production ↓	Animal study: i.p./oral/topical treatment
Senicapoc	KCa3.1	immune cells	KCa3.1-selective blocker	inflammation ↓	Animal study: oral intake

**Table 3 ijms-25-02756-t003:** The information regarding the use of drugs and clinical trials listed in Table 2 is provided.

Drugs	Main Use of Drugs	FDA Approved	Target Diseases of Clinical Trial (Referenced from clinicaltrials.gov (accessed on 13 February 2024))
Tropisetron	antiemetics	O	
Tacrolimus *	immunosuppressive drug	O	
Cyclosporine *	immunosuppressive drug	O	
Dexamethasone *	glucocorticoid medication	O	
Resveratrol *	nutritional supplement		
Anandamide			migraine [154]
Xanthotoxin * (methoxsalen)	psoriasis [155], eczema, vitiligo, and some cutaneous lymphomas	O	
Thymol *	essential oil compound		
Erlotinib	anti-cancer	O	
LY294002			schizophrenia anti-cancer
Baicalein			anti-inflammatory
Glibenclamide	hypoglycemic agents	O	
Stichodactyla helianthis neurotoxin (Analogue: ShK-186 (Dalazatide))			plaque psoriasis [156]
Senicapoc			respiratory diseases, sickle cell anemia

* Approved for use on the human skin or available for use.

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
