# Peer review of "Pathophysiological Roles of Ion Channels in Epidermal Cells, Immune Cells, and Sensory Neurons in Psoriasis"

_ijms, 2024, doi:10.3390/ijms25052756_

Round 1
Reviewer 1 Report
Comments and Suggestions for Authors
This review describe in detail the ion channels potentially involved in the interplay among keratinocytes, immune cells and neurons in psoriasis.
The review is well written and presented. Although the topic is complex and intricate, the authors give an enough clear scenario of what can happen in psoriasis.
The only point is to put the table I before the last paragraph, as well as the table 2. The table I list the different ion channels present on the major actors of psoriasis. I would insert this at the end of the introduction (this can help the reader to see before the description what will be considered in the manuscript). The table 2 just after the point 4.4.
Also, the abstract can be more detailed. In this present form, no specific ion channels are considered. The authors can list the major ion channels important in psoriasis.
Comments on the Quality of English Language
English language is good.
Author Response
Reviewer 1
This review describe in detail the ion channels potentially involved in the interplay among keratinocytes, immune cells and neurons in psoriasis. The review is well written and presented. Although the topic is complex and intricate, the authors give an enough clear scenario of what can happen in psoriasis.
[Response] First of all, we really appreciate the reviewer’s constructive comments. In the current revised manuscript, we have addressed each issue raised by the reviewer on a point-by-point basis. Our answers to the questions are listed below, and the main text file has been modified according to the valuable comments of the referee. All revisions to the manuscript are highlighted in red.
- The only point is to put the table I before the last paragraph, as well as the table 2. The table I list the different ion channels present on the major actors of psoriasis. I would insert this at the end of the introduction (this can help the reader to see before the description what will be considered in the manuscript). The table 2 just after the point 4.4.
[Response] As the reviewer suggested, we have rearranged the positioning of Table 1 and Table 2 in the revised manuscript.
- Also, the abstract can be more detailed. In this present form, no specific ion channels are considered. The authors can list the major ion channels important in psoriasis.
[Response] We have revised the abstract of the revised manuscript based on the reviewer’s comment (Line 30~32).

Reviewer 2 Report
Comments and Suggestions for Authors
The comments attached

Author Response
Reviewer 2
The authors have well described pathophysiological roles of ion channels in epidermal cells, immune cells, and sensory neurons in the chronic disease of psoriasis in this review article. In addition, the authors expected to provide further details in this manuscript. The author should incorporate the editing prior to submission of this manuscript.
[Response] First of all, we really appreciate the reviewer’s constructive comments. In the current revised manuscript, we have addressed each issue raised by the reviewer on a point-by-point basis. Our answers to the questions are listed below, and the main text file has been modified according to the valuable comments of the referee. All revisions to the manuscript are highlighted in red.
- The authors have provided cartoon sketch figures 2 and 3 for ion-channel signaling pathway figures for keratinocyte and sensory neurons, in similar way the additional figure should be provided for immune cells
[Response] As the reviewer suggested, we have further included Figure 3 illustrating the functional ion channels in immune cells in the revised manuscript.
- Table three for the drug candidate action is confusing and this should be divided into three categories and in case any drug candidate works for multiple channel groups then further should be provided in additional graphical presentations.
[Response] In the revised manuscript, the table pointed out by the reviewer has been divided into three categories, comprising epidermal cells, immune cells, and sensory neurons. In addition, there were no drug candidates identified for treating psoriasis that were found to work across multiple channel groups.
- From table 3 how many drugs are in clinical trial and/or FDA approved for clinical practice need to be listed in separate table.
[Response] In the original manuscript, there were Tables 1 and 2. Regarding the drugs listed in Table 2, a summary of those undergoing clinical trials or approved by the FDA has been compiled and incorporated into Table 3 in the revised manuscript.
- Does any drug candidate FDA approved for its clinical application the highlighted them as wells by showing repurposing of the drug candidate.
[Response] As the reviewer suggested, we have also included information on drug repurposing of FDA-approved drugs in Table 3.
- The authors described the topical application of some drugs like capsaicin and PAP1 which need to provide the information regarding skin penetration and potential interaction with blood vessels/ capillaries, immune cells, and sensory nerve fiber endings.
[Response] As the reviewer suggested, specific information on topical application was included in the revised manuscript. Moreover, additional details on the effects on blood vessels, side effects, and skin effects were provided (Line 676~689).
- The perspectives and conclusion should be placed after table 2 as a last part of this manuscript.
[Response] Following the reviewer's suggestion, we have reorganized the placement of the perspectives and conclusion section after Table 2 in the revised manuscript.

Reviewer 3 Report
Comments and Suggestions for Authors
In the present article, the authors provide an extensive review of the role of ion channels in psoriasis. Psoriasis is a common inflammatory skin disease that affects 2-3% of the population. Traditionally, psoriasis is thought to be a skin disorder, however, psoriasis has effects on the peripheral nervous system, involving ion channels that control itch, pain, and immune responses. Further, ion channels affect various processes in keratinocytes, immune cells, and neurons that in turn form a complex cellular network.
More specifically, the review focuses on nicotinic acetylcholine receptors (nAchRs) and transient receptor potential (TRP) channels in keratinocytes and their roles in the epidermal barrier function and in immune cells. The potential therapeutic implications of targeting these ion channels for psoriasis treatment are highlighted.
The authors have gathered important information mainly clinical data (both expression profiles as well as potential links between mutations in ion channels and psoriasis phenotype such as gain-of-function mutations in TRPV3 are associated with increased itching) and from animal models of psoriasis (IMQ treated mice) and genetically engineered mice (e.g. a7nAChR deficient mice) and have combined all these data to construct their article.
Overall, the article is interesting and highlights the role of ion channels in the development of psoriasis that opens new ways to treat psoriasis in combination or alternatively to the new methods that are currently available (mainly immunotherapy).
Therefore I recommend the acceptance of this article without any additional comments.
Author Response
Reviewer 3
In the present article, the authors provide an extensive review of the role of ion channels in psoriasis. Psoriasis is a common inflammatory skin disease that affects 2-3% of the population. Traditionally, psoriasis is thought to be a skin disorder, however, psoriasis has effects on the peripheral nervous system, involving ion channels that control itch, pain, and immune responses. Further, ion channels affect various processes in keratinocytes, immune cells, and neurons that in turn form a complex cellular network. More specifically, the review focuses on nicotinic acetylcholine receptors (nAchRs) and transient receptor potential (TRP) channels in keratinocytes and their roles in the epidermal barrier function and in immune cells. The potential therapeutic implications of targeting these ion channels for psoriasis treatment are highlighted. The authors have gathered important information mainly clinical data (both expression profiles as well as potential links between mutations in ion channels and psoriasis phenotype such as gain-of-function mutations in TRPV3 are associated with increased itching) and from animal models of psoriasis (IMQ treated mice) and genetically engineered mice (e.g. a7nAChR deficient mice) and have combined all these data to construct their article. Overall, the article is interesting and highlights the role of ion channels in the development of psoriasis that opens new ways to treat psoriasis in combination or alternatively to the new methods that are currently available (mainly immunotherapy).
Therefore, I recommend the acceptance of this article without any additional comments.
[Response] we really appreciate the reviewer’s positive comment.

Reviewer 4 Report
Comments and Suggestions for Authors
The content of this review was to report the pathophysiological roles of ion channels in epidermal and immune cells, and sensory neurons in psoriasis. In section of Perspectives, the authors emphasized that the biological mechanisms of ion channels involved in psoriasis and their role.”. Table 1 and 2 listed related literature. Please supply the quantitate criterion about this statement. What is the positive percentage of the statement and what is the negative percentage of this statement. Then explain their idea for further study.
Comments on the Quality of English LanguageModerate editing of English language required
Author Response
Reviewer 4
The content of this review was to report the pathophysiological roles of ion channels in epidermal and immune cells, and sensory neurons in psoriasis. In section of Perspectives, the authors emphasized that the biological mechanisms of ion channels involved in psoriasis and their role.”. Table 1 and 2 listed related literature. Please supply the quantitate criterion about this statement. What is the positive percentage of the statement and what is the negative percentage of this statement. Then explain their idea for further study.
[Response] First of all, we really appreciate the reviewer’s constructive comments. In the current revised manuscript, we have addressed each issue raised by the reviewer on a point-by-point basis. Our answers to the questions are listed below, and the main text file has been modified according to the valuable comments of the referee. All revisions to the manuscript are highlighted in red.
For the reviewer’s comments, we have provided further clarification on the positive and negative functions of ion channels in psoriasis pathogenesis in the revised manuscript. Please inform us if we have misunderstood your comments.
Several ion channels function as positive triggers for psoriasis. Overexpression, gain of function, and activation of channels, including nAchR, TRPV1, TRPV3, TRPV4, TRPM4, and ANO1 in keratinocytes, as well as TRPV1, nAchR, Kv1.3, and KCa3.1 in immune cells, have been associated with the induction of psoriasis. Thus, inhibitors directed at these channels could be potential candidates for therapeutic intervention. On the contrary, polycystin, CFTR, TRPC, and CRAC channels in keratinocytes exert repressive roles on psoriasis development. Hence, agonists targeting these ion channels in keratinocytes could be potential substances for psoriasis treatment.
Interestingly, the TRPA1 channel appears to have dual functions—both positive and negative—in relation to psoriasis pathogenesis, depending on the cell type expressing it. Activation of TRPA1 in keratinocytes is implicated in the development of psoriasis, while its activation in immune cells is speculated to have suppressive effects on psoriasis pathogenesis. Additionally, it's noteworthy that among nAchR types, alpha5 exhibits pathological effects on psoriasis, whereas alpha7 shows a negative impact (alpha7 agonists have been shown to alleviate psoriasis symptoms).
Therefore, selectively modulating specific ion channel types or subunits can indeed lead to more effective results. Additionally, understanding the interaction between different cell types, such as keratinocytes, immune cells, and sensory neurons, is essential for developing targeted therapies. Research efforts should focus on identifying ways to target ion channels within these specific cell types to tailor treatments that address the intricate dynamics of psoriasis pathogenesis. This comprehensive approach could pave the way for more precise and efficient interventions in the management of psoriasis.
This discussion has added in the Perspectives section in the revised manuscript (Line 871~892).
